# Deep learning-based segmentation of rabbit fetal skull with limited and sub-optimal training labels

**Rajath Soans**[1]     SOANS@MERCK.COM
**Alexa Gleason**[1]     ALEXA_GLEASON@MERCK.COM
**Tosha Shah**[1]     TOSHA_SHAH@MERCK.COM
**Corey Miller**[1]     CORIN_MILLER@MERCK.COM
**Barbara Robinson**[1]     BARBARA_ROBINSON@MERCK.COM
**Kimberly Brannen**[1]     KIMBERLY.BRANNEN@MERCK.COM
**Antong Chen**[1]     ANTONG.CHEN@MERCK.COM

[1] *Merck & Co., Inc., Rahway, NJ, USA*

## Abstract

In this paper, we propose a deep learning-based method to segment the skeletal structures in the micro-CT images of Dutch-Belted rabbit fetuses which can assist in the detection of skeletal abnormalities in nonclinical developmental toxicity studies. Our strategy leverages sub-optimal segmentation labels of 22 skull bones from 26 micro-CT volumes and maps them to 250 unlabeled volumes on which a deep CNN-based segmentation model is trained. In the experiments, our model was able to achieve an average Dice Similarity Coefficient (DSC) of 0.89 across all bones on the testing set, and 14 out of the 26 skull bones reached average DSC >0.93. Our next steps are segmenting the whole body followed by developing a model to identtify abnormalities.

**Keywords:** U-Net, nonclinical drug safety assessment, DART, micro-CT, rabbit fetus, sub-optimal ground truth training label, sparse label map

## 1. Introduction

A common component of nonclinical safety assessments for new pharmaceuticals is the evaluation of potential effects on prenatal development, including an assessment of fetal skeletal development, most often in rats and rabbits. This assessment is usually accomplished by visual inspection of a specimen stained with Alizarin Red S, but alternative methods to use micro-computed tomography (CT) with inspection of a 3-D reconstructed image have been developed (example shown in Figure 1) (Winkelmann and Wise, 2009)

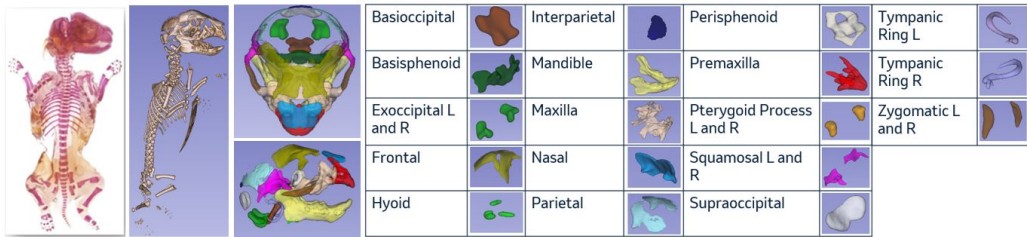

Figure 1: **Dutch-belted rabbit fetus.**(left to right) alizarin red staining; Rendering of the skeletal structure from micro-CT image; Color coded label maps; table illustrating 22 bone segments of the skull

Automation of such processes would require segmentation of each bone from the skeleton, however, training a segmentation model is challenged by a) lack of annotated data

and b) sub-optimal quality of annotations (Tajbakhsh et al., 2020). Acquiring sufficient and accurate manual annotations on complicated skeletal structures is expensive and impractical. In our work, we leverage annotations that are poorly delineated and available only in a limited quantity. We use image registration to map these annotations to a larger dataset which is then used to train a deep convolutional neural network (CNN) to perform automated segmentation.

## 2. Materials and Methods

Micro-CT images were acquired using GE Locus Ultra micro-CT scanner with a polystyrene holder bucket containing up to 9 rabbit fetuses in each scan. Image volumes were reconstructed with voxel size of $0.1 \times 0.1 \times 0.1 \ mm^3$ and scaled to Hounsfield units (HU).

To analyze the fetus skull, we first cropped a sub-volume of size $320 \times 320 \times 250$ containing the skull region with 250 slices on the z-direction. From a legacy set of 513 volumes segmented using a previously proposed automated segmentation pipeline (Dogdas et al., 2015), although the segmentation labels are sub-optimal, we inspected them and selected 26 volumes with relatively more accurate and complete segmentation labels to be the atlases for a multi-atlas segmentation (MAS) strategy shown in Figure 2.

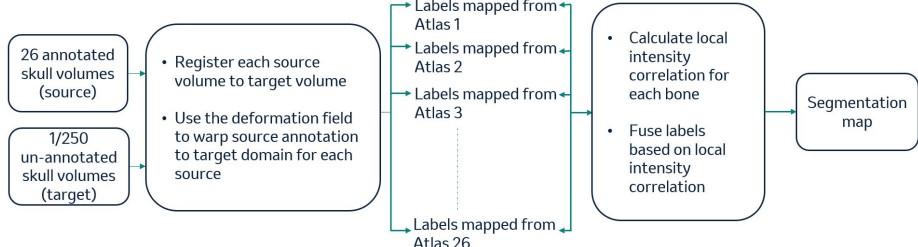

Figure 2: **MAS workflow.** Registration from source to target are performed in this order- *global rigid* $\rightarrow$ *global non-rigid* $\rightarrow$ *local non-rigid* using ANTS suite (Avants et al., 2009). Fusion weights are proportional to local intensity correlation.

Although the MAS strategy is effective, the execution of the registration workflow is time consuming and can absorb substantial amount of computing resource. Therefore, we elect to leverage the MAS strategy to create a dataset to train a U-Net segmentation model (Ronneberger et al., 2015). Specifically, the MAS strategy is used in obtaining segmentation maps for a set of 250 un-annotated images which is then partitioned into 220 training and 30 testing images. The segmentation maps representing just a single bone segment tend to pose difficulty in training due to its sparse nature. To overcome this challenge, we obtained distance transform of the segmentation maps and used it in guiding the model to convergence. This was realized by designing the loss function using a combination of a normalized distance regression loss (Ma et al., 2020) and the Dice Similarity Co-efficient (DSC) as shown in 1.

$$L = \alpha * Dice\_loss + \beta * \frac{1}{N|\Omega|} \sum_{\Omega} SDM(ground\_truth) \ o \ predicted\_map \qquad (1)$$

where $SDM()$ is the function to obtain Signed Distance Map as defined in (Xue et al., 2020), $N$ is the normalization factor to scale both losses to the same range, $\Omega$ is the grid on which

image is defined, $\alpha$ and $\beta$ are the co-efficients, and $o$ is the Hadamard product. Training is initialized with higher $\beta$ (0.8 in our experiments) and every 10 epochs it is reduced by 10% with an equal increase in $\alpha$. Our overall pipeline is illustrated in Figure 3.

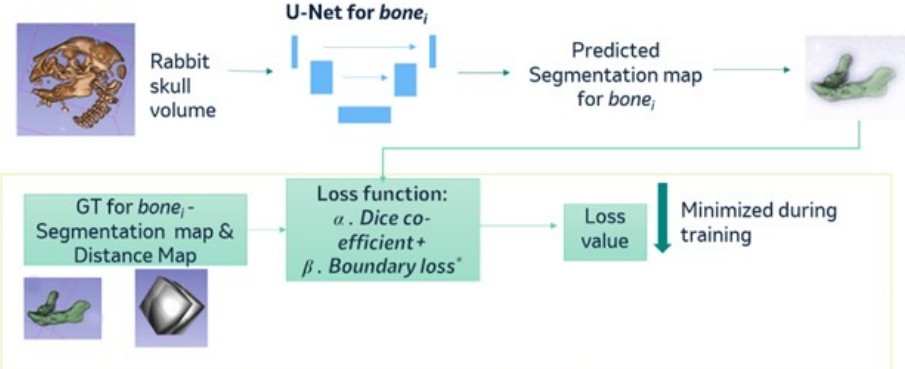

Figure 3: **U-Net segmentation pipeline.** 22 models are trained targeting one bone segment per model.

## 3. Results and Conclusion

DSC profile for U-Net segmentations on 30 test images is shown in the left panel of Figure 4. To make an intuitive assessment, we used the U-Net based approach to regenerate segmentations on the original 26 atlases to compare with the sub-optimal ground truth labels. Example cases are shown in the right panel of Figure 4.

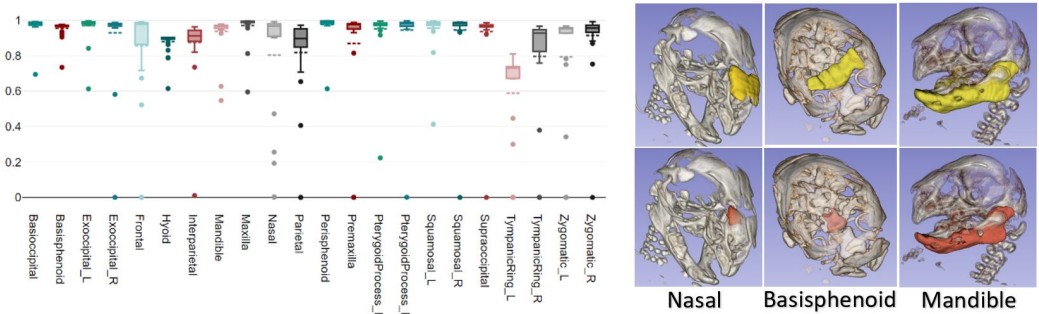

Figure 4: **U-Net segmentation results. (left) DSC boxplot and (right) visualization.** U-Net predictions has a DSC >0.9 for most bones. Smaller and thinner bones e.g. Tympanic Rings are challenging to segment, yielding low DSC. Example segmentations on the original 26 atlases (yellow: U-Net predictions (top row); red: ground truth (bottom row)) illustrating improvement over the ground truth on the atlases, showing robustness of our MAS+U-Net based approach and its ability to overcome sub-optimal labels.

Our proposed segmentation strategy is effective and can function as the initial step in identifying anomalies in rabbit fetus skull bones. We will further explore segmentation of the whole body skeleton which is relatively more challenging due to higher degree of inter-specimen variability.

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
