# OpenReview forum: "Deep learning-based segmentation of rabbit fetal skull with limited and sub-optimal training labels"
_MIDL.io/2023/Short_Paper_Track — MIDL 2023 Short paper track Poster_

### Official Review · Reviewer_xydm · 2023-04-15
**nice paper, good technical novelty**

**Rating:** 8
**Confidence:** 4

**Review:**

nicely written paper --clear exposition

Interesting two-step approach: multi-atlas segmentation followed by UNet modeling, also using distance maps in loss function (technical novelty)

---

### Official Review · Reviewer_58wj · 2023-04-22

**Rating:** 4
**Confidence:** 5

**Review:**

This work presents a workflow for training a Unet to segment skull bones in micro-CT images of rabbit fetuses. This is done with the following steps: In a previous study, an automated method had segmented 513 volumes. Via visual inspection, the authors chose 26 of these volumes that were deemed as "relatively more accurate and complete". Then, 250 unlabelled images are segmented, by using Multi-Atlas-Segmentation (MAS) via registration, registering the 26 segmentations (atlases) to each of the 250 unlabelled images and fusing the propagated segmentations. This framework, although the authors claim it gives good segmentations, it is time consuming (registraiton is slow). Therefore, the authors then set of to train a Unet on these MAS segmentations, as the Unet can then produce segmentations fast. For this purpose, these 250 segmentations made by MAS are divided into 220 for training the Unet, and 30 for testing/evaluating the Unet. The Unet achieves high DSC (>90%) on the 30 test-images.

Strengths:
- A tool for automatic segmentation of fetuses of rabbits could help studies on development.
- The paper is well written.

Weaknesses:
- Technical novelty is limited as the work is a standard workflow, of training a Unet on automatically created segmentations (by MAS).
- The evaluation is not on manual segmentations, but on segmentations produced by Multi-Atlas-Segmentation (label propagation via registration). The quality of these segmentations by MAS has not been analysed. As a results,  it is not possible to estimate how good is the segmentation made by MAS that is used as the golden truth for calculation of DSC to evaluate the Unet. Therefore, the reader really cannot conclude whether the resulting model is actually good, or simply achieves high DSC by replicating the same mistakes that MAS does.
- The quality of the original 26 labels that are the original input to the workflow, has not been analysed. Therefore, I don't think the current work can support the claim made that the framework learns from "poor" annotations (as per title and text). In fact the authors claim they have been chosen after visual inspection, and chosen because they were "relatively more accurate and complete". How accurate / how poor? There is no analysis and therefore no conclusion can be made.

For the above, I dont think I can extract useful conclusions from the work in its current state.

Comments for future extension / improvements:
I think such a tool could be very useful in the field. I would recommend the authors to try and think how they could carefully evaluate the quality of the workflow. E.g. by creating some manual segmentations for evaluation (even a few), or conduct some analysis to quantify how good/bad the segmentations that are used (original 26 and/or MAS) are, since they are used to make the central claims of the work.